# COVID-19 Vaccine-Related Vogt–Koyanagi–Harada Disease Complicated by Central Serous Chorioretinopathy during Treatment Course: Case Report and Literature Review

**DOI:** 10.3390/vaccines10111792

**Published:** 2022-10-25

**Authors:** Ruyi Han, Gezhi Xu, Xinyi Ding

**Affiliations:** 1Department of Ophthalmology, Eye and ENT Hospital of Fudan University, Shanghai 200031, China; 2Shanghai Key Laboratory of Visual Impairment and Restoration, Fudan University, Shanghai 200031, China; 3Key Laboratory of Myopia of National Health Commission, Fudan University, Shanghai 200031, China; 4Key Laboratory of Myopia, Chinese Academy of Medical Sciences, Shanghai 200031, China

**Keywords:** COVID-19, vaccination, ocular complication, Vogt–Koyanagi–Harada disease, central serous chorioretinopathy disease

## Abstract

With the promotion of mass COVID-19 vaccination in the elimination of the SARS-CoV-2 pandemic, new side effects, including ocular complications, are emerging. In this study, we report on a 62-year-old Chinese man who developed Vogt–Koyanagi–Harada (VKH) disease six days after his third dose of an inactivated COVID-19 vaccine, with a preceding severe headache and tinnitus. His medical history included tuberculosis 20 years prior and hypertension. Systemic prednisone was administered, resulting in completely relieved inflammation and improved visual acuity. Another three and a half months later, the visual acuity of his right eye slightly decreased due to complicated central serous chorioretinopathy (CSC) disease. By gradually replacing prednisone with cyclosporine within 2 months, the subretinal fluid was completely absorbed at the last visit. Steroid-related CSC during the treatment course of VKH disease after COVID-19 vaccination has never been reported before. By reviewing relative literature, we discuss the mechanism of CSC onset in our case and the potential therapeutic strategies. Complicated CSC may develop in the eyes with vaccine-related VKH after steroid treatment. Ophthalmologists should be aware of this condition, carefully distinguish complicated CSC with inflammation relapse, and adjust the medication in a timely manner.

## 1. Introduction

Vaccination against coronavirus disease 2019 (COVID-19) has been deployed in the elimination of the SARS-CoV-2 pandemic, providing significant protection against the infection and the development of acute respiratory distress syndrome and multi-organ failure [1]. However, emerging evidence has demonstrated potential adverse events in vaccinated patients, including ocular complications. Various ocular adverse events, including non-arteritic anterior ischemic optic neuropathy, central serous chorioretinopathy (CSC), and Vogt–Koyanagi–Harada (VKH) disease, have been reported [2,3,4,5]. VKH disease is a chronic, bilateral, granulomatous ocular and multisystem inflammatory disorder [6] that responds well to adequate, long-range steroid therapy with or without immunosuppressive agents [3,4,7,8,9,10]. Here, we report on a patient who developed an incomplete VKH disease in close temporal association with an inactivated COVID-19 vaccine. However, during the course of steroid treatment, CSC developed in one of his eyes, which changed our treatment strategy.

## 2. Case Report

Six days after receiving a third dose of an inactivated COVID-19 vaccine (Sinopharm (Vero cell), inactivated COVID-19 vaccine) on 24 November 2021, a 62-year-old Chinese man developed a severe headache and tinnitus and subsequently suffered bilateral vision decline. He presented at our uveitis specialist clinic on 18 January 2022 and reported a history of bilateral iridocyclitis diagnosed on 22 December 2021. Topical prednisolone acetate (q2h), pranoprofen (qid), and atropine (tid) for both eyes were provided by the previous ophthalmologist. Other medical history included cured tuberculosis (TB) 20 years prior and hypertension. He had no history of penetrating ocular trauma or surgery preceding the initial onset of uveitis.

In the ophthalmological evaluation, his best-corrected visual acuity (BCVA) was 20/100 in OD and 20/33 in OS. Examination under a slit lamp revealed a clear cornea and anterior chamber and posterior subretinal fluid OU. Conventional ultrasonography (Aviso, Quantel Medical, Cournon d’Auvergne Cedex, France) and enhanced-depth imaging swept-source optical coherence tomography (EDI-OCT, Spectralis OCT; Heidelberg Engineering, Heidelberg, Germany) revealed serous retinal detachment (SRD) and significant choroidal thickening in both eyes (Figure 1a). Swept-source optical coherence tomography angiography (SS-OCTA, PLEX Elite 9000^®^device, Carl Zeiss Meditec, Dublin, CA, USA) exhibited a distributed flow void in the choriocapillaris layer OU. Blood work, including a test for human immunodeficiency virus (HIV), a Treponema pallidum particle assay (TPPA), a rapid plasma reagin (RPR) test, and a test for autoinflammation markers, was unremarkable. The interferon gamma release assay was positive, and the chest CT showed old pulmonary nodules. Considering that the patient had accepted regular anti-tuberculosis treatment previously, no other medical treatment against TB except for close observation and chest CT re-examination 6 months later was suggested by lung specialists.

Based on his medical history, neurological findings, and ocular findings, we diagnosed incomplete Vogt–Koyanagi–Harada disease. Oral prednisone (50 mg, qd) was used as an initial treatment [6]. Additionally, the close temporal association between the COVID-19 vaccination and the initial presentation of VKH disease suggested that the vaccine against SARS-CoV-2 could have been the trigger for VKH disease in this case. According to the Naranjo adverse drug reaction probability scale (Table 1), a score of four indicates that the observed symptoms were a possible adverse drug reaction [11]. One week later (26 January 2022), his BCVA had improved to 20/33 in OD and 20/22 in OS. Follow-up EDI SS-OCT demonstrated a significant reduction in subretinal fluid and relief of choroid edema OU (Figure 1b). One month after treatment (25 February 2022), his BCVA had steadily improved to 20/22 in OD and 20/20 in OS. OCT demonstrated very little subretinal fluid in the right eye and a total recovery of retinal structure in the left eye (Figure 1c). Oral prednisone administration was gradually tapered.

The patient did not show up until 3 and a half months later (7 June 2022) due to the COVID-19 pandemic in Shanghai. His oral prednisone dose had been tapered to 20 mg (qd). He complained about slightly blurred vision in both eyes (20/25 in OD and 20/22 in OS). A slit lamp examination presented few cells in vitreous body OU and sub-macular fluid OD. EDI-OCT showed sub-macular fluid OD and an increase in choroid thickness OU (Figure 2a). The condition was considered a relapse of inflammation. Thus, the patient was given 30 mg (qd) of oral prednisone and 125 mg (bid) of cyclosporin. Improved BCVA, 20/22 in OD and 20/20 in OS, and clear vitreous body a week later indicated the relief of inflammation. However, in the following observations (1 July 2022 and 29 July 2022), the retinal fluid in his right eye gradually increased (Figure 2b,c) without any sign of activated inflammation and quiet anterior and posterior segments were observed in his left eye. A dilated fundus examination revealed a depigmented fundus appearance and sub-macular fluid in his right eye (Figure 3a). Indocyanine green angiography (ICGA) (Heidelberg Engineering, Heidelberg, Germany) of the right eye showed the focal dilation of choroidal veins and delayed perfusion in the early phase (Figure 3d). Hyperfluorescent spots and patches with slight leakage were superior to the fovea, which showed strengthened fluorescence in the late phase. Fluorescein angiography (FA) (Heidelberg Engineering, Heidelberg, Germany) demonstrated that the hyperfluorescent patch superior to the fovea gradually leaked over time and displayed strengthened fluorescence in the late phase (Figure 3d). OCT revealed subretinal fluid in fovea and the irregular RPE, suspected RPE break, and the elongation of photoreceptor cells segments in the region of serous retinal detachment (Figure 3b,c). Complicated central serous chorioretinopathy (CSC) in his right eye was diagnosed, and the tapering of oral prednisone by 5 mg a week was set. Additionally, the patient was given one micropulse laser treatment. One and a half months later (13 September 2022), subretinal fluid was completely absorbed (Figure 2d).

## 3. Discussion

In this paper, we report that a vaccinated patient developed complicated CSC during steroid treatment for vaccine-associated VKH disease. After we replaced oral prednisone with cyclosporin, the SRD in his eye was significantly relieved.

VKH disease closely related to vaccines against SARS-CoV-2 is a quite rare event, although it has been recognized [3,4,8,9,12,13,14,15]. Papasavvas et al. reported the reactivation of well-controlled VKH disease for more than 6 years following an anti-SARS-CoV-2 vaccination [12]. Saraceno et al. first described a patient diagnosed with complete VKH syndrome 4 days after receiving COVID-19 immunization with the Oxford–AstraZeneca Chimpanzee Adenovirus Vectored Vaccine ChAdOx1 nCoV-19 (AZD1222) [13]. In a previous study, we also reported that a 33-year-old Chinese man developed probable VKH disease only one day after his first dose of an inactivated COVID-19 vaccine and summarized the clinical characteristics of VKH disease in temporal association with COVID-19 vaccination [4]. Different possibilities regarding the causality of VKH and vaccination have been raised, including adjuvants [13], the antigen mimicry [8], and structural surface glycoprotein antigen [12]. However, the precise mechanism of this phenomenon remains undetermined.

The therapeutic strategy for vaccine-related VKH disease was the same as that for regular VKH disease. Oral steroids are generally used in VKH disease treatment and have achieved good responses in the majority of vaccine-associated VKH patients [16,17,18]. In our case, the subretinal fluid in both eyes was significantly absorbed in the first month of treatment with oral prednisone, and visual acuity was steadily improved. Immunosuppressive agents have been employed to suppress the ocular inflammation in patients who present in the chronic phase or with chronic recurrent disease, and these agents have proven to be a great complement for corticosteroids in VKH disease therapy [19]. In combination with steroids, immunosuppressive agents have also been used in vaccinated patients with VKH disease [12,20]. A previous study demonstrated that in combination with low doses of corticosteroids, cyclosporine A can effectively decrease IL-17 and IFN-γ, while prednisone does not completely inhibit the production of IL-17 and IFN-γ in VKH disease [21]. Compared with the treatment outcomes of patients treated with prolonged steroid therapy with or without the delayed addition of immunosuppressive agents, patients initially treated with immunosuppressive agents have shown better visual outcomes [22,23,24]. However, well-controlled, prospective, randomized clinical trials are needed to find out whether immunosuppressive agents should be employed as a first-line therapy for VKH disease.

Recurrent VKH disease and complicated CSC should be considered when subretinal fluid is increased during the treatment of VKH disease. Eyes with recurrent VKH disease may present with an active ocular inflammation response and respond well to steroid treatment, while eyes with complicated CSC have no ocular inflammation response and may be aggravated by steroid treatment. Although eyes with CSC or recurrent VKH disease are both characterized by thick choroids, choroidal folds and blurry choroidal vessel structures only present in eyes with recurrent VKH disease [16]. In the sixth month of steroid treatment for VKH disease in our case, the subretinal fluid and choroid thickness were significantly increased in the right eye of the patient and a few cells in vitreous body OU were observed. This was originally considered a relapse of inflammation, and additional dosages of steroids and cyclosporin were provided. However, during the following visits, the vitreous body of the patient’s right eye became clear while the subretinal fluid was continuously increased. OCT revealed a suspected RPE break and clear but dilated choroid vessels underneath the region of subretinal fluid. ICG and FA demonstrated the focal dilation of choroidal veins and hyperfluorescent spots and patches superior to the fovea, which gradually leaked over time and displayed strengthened fluorescence in the late phase. Based on these observations, complicated CSC in his right eye was diagnosed. So far, this is the first case demonstrating steroid-related CSC during the treatment course of VKH disease associated with COVID-19 vaccination.

Numerous risk factors have been outlined in developing CSC, including the use of local and systemic steroids [25,26,27,28,29,30]. The high-dose use of corticosteroids was suggested to damage the posterior blood–ocular barrier [25], though the precise mechanisms are largely unknown. Recently, Zhao et al. provided evidence demonstrating that glucocorticoids may bind the mineralocorticoid receptor (MR) and activate the MR signaling pathway, resulting in choroidal enlargement [31]. Treated with a specific MR antagonist, patients with chronic non-resolved CSC have shown the impressive and rapid resolution of retinal detachment and choroidal vasodilation, as well as improved visual acuity [31]. However, one study showed that steroid-induced CSC may be an idiosyncratic response in selected vulnerable individuals rather than a dose-dependent effect [32]. The researchers measured the choroid thickness in 18 patients systemically treated with a high dose of corticosteroid (19.5 ± 4.1 mg/kg per day). The mean subfoveal choroidal thickness at baseline was 259.8 μm and showed no significant change at 1 day, 1 week, or 1 month after steroid administration. Only one patient who had presented with pigment epithelial detachment and a thick choroid (381.1 μm) developed bilateral CSC. In our case, after the complete resolution of active inflammation, the choroid thickness of the patient’s right eye was still as high as 393 μm (25 February 2022), which could be considered a pachychoroid. After another 3-month treatment of oral steroids, the choroidal thickness of his right eye was increased to 464 μm and the choroidal thickness of his left eye was also slightly increased (from 326 to 336 μm), which might be explained by MR signaling pathway activation [31]. Thus, we consider that pachychoroid and systemic steroids might have together contributed to the development of complicated CSC in our case. We recommend that ophthalmologists pay attention to the possible ocular side effects of systemic steroids, especially in eyes with a pachychoroid.

The therapeutic strategy should be changed in a timely manner when CSC occurs during the course of steroid treatment. Due to the high rate of spontaneous subretinal fluid resolution within three to four months in steroid-related CSC, the observation and elimination of risk factors are preferred in the relevant treatment, which should include the discontinuation of the use of all forms of corticosteroids [33,34]. ICGA- and FA-guided half-dose PDT, laser photocoagulation on focal leakage points, micropulse laser treatment, and MR antagonists also show benefits in decreasing the time to restore vision [35,36,37]. In our case, after the diagnosis of complicated CSC, oral prednisone was replaced by immunosuppressants and micropulse laser treatment was employed once. One and a half months after the withdrawal of oral steroids, the subretinal fluid was completely absorbed, which further suggested the steroids as a possible triggering factor of CSC in this case. In addition, the use of immunosuppressants at an appropriate dose is vital to prevent the recurrence of VKH inflammation during the withdrawal of steroids.

## 4. Conclusions

VKH disease is an immune-mediated bilateral uveitis that is associated with neurological, auditory, and integumentary manifestations. Triggering factors include microbial infection, virus DNA, and vaccines. With the promotion of COVID-19 vaccination, vaccine-related VKH disease has been reported. Systemic steroids are a mainstay in the therapy of VKH disease and result in good responses in vaccine-related VKH patients. However, systemic steroids may induce CSC, especially in patients who have thick choroids. Here, we have reported complicated CSC during the steroid treatment course of vaccine-related VKH and discussed the possible mechanisms of CSC onset. Although acute CSC is self-limited and resolution can be observed with the tapering of steroids in most cases, physicians and ophthalmologists should be aware of this condition (especially in eyes with a pachychoroid), carefully distinguish complicated CSC with inflammation relapse, and adjust the medication in a timely manner.

## Figures and Tables

**Figure 1 vaccines-10-01792-f001:**
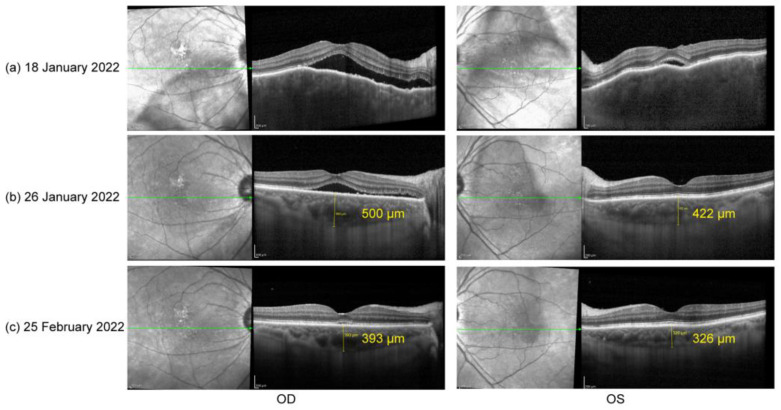
(**a**) Optical coherence tomography (OCT) on 18 January 2022: serous retinal detachment (SRD) and choroidal thickening OU; (**b**) OCT on 26 January 2022: significant reduction in subretinal fluid and relief of choroid edema OU; (**c**) OCT on 25 February 2022: little subretinal fluid in OD and a total recover of retinal structure in OS.

**Figure 2 vaccines-10-01792-f002:**
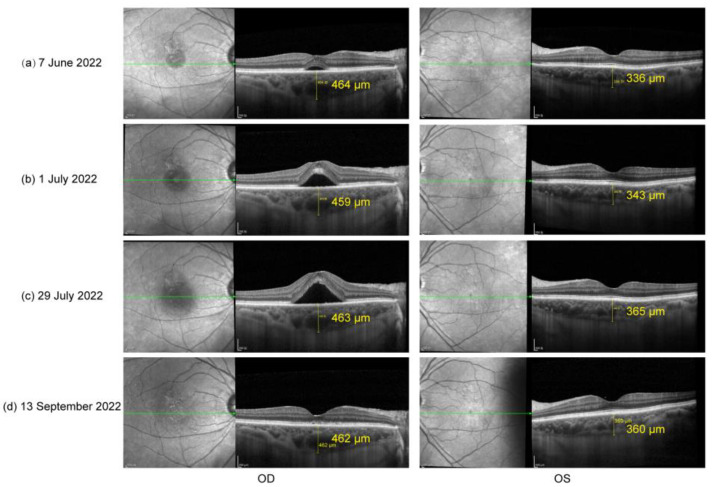
(**a**) OCT on 7 June 2022: sub-macular fluid OD and increase in choroid thickness OU; (**b**) OCT on 1 July 2022 and (**c**) OCT on 29 July 2022: significantly increased sub-macular fluid OD; (**d**) OCT on 13 September 2022: completely absorbed subretinal fluid.

**Figure 3 vaccines-10-01792-f003:**
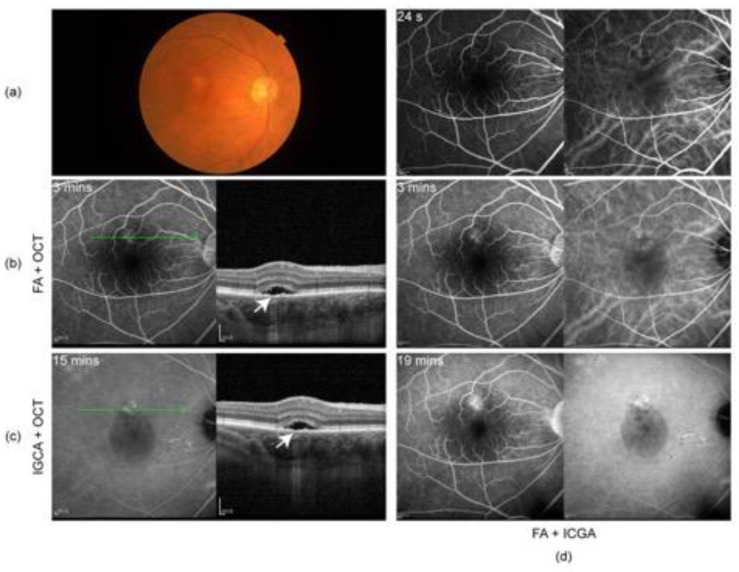
(**a**) Color fundus photography on 29 July 2022 showing mild depigmented fundus appearance and sub-macular fluid; (**b**) fluorescein angiography (FA) and OCT and (**c**) indocyanine green angiography (ICGA) and OCT on 29 July 2022: subretinal fluid in fovea and irregular RPE, suspected RPE break (arrow), and elongation of photoreceptor cells segments in the area of SRD; (**d**) ICGA and FA on 29 July 2022: focal dilation of choroidal veins and delayed perfusion in the early phase, hyperfluorescent spots, and patches with slight leakage in superior to the fovea, showing strengthened fluorescence in the late phase.

**Table 1 vaccines-10-01792-t001:** Naranjo adverse drug reaction (ADR) probability scale.

ADR Probability Scale					
To Assess the Adverse Drug Reaction, Please Answer the Following Questionnaire and Give the Pertinent Score					
		Yes	No	Do not know	Score
1	Are there previous conclusive reports on this reaction?	+1	0	0	0
2	Did the adverse event appear after the suspected drug was administered?	+2	−1	0	2
3	Did the adverse reaction improve when the drug was discontinued or a specific antagonist was administered?	+1	0	0	0
4	Did the adverse reaction reappear when the drug was readministered?	+2	−1	0	0
5	Are there alternative causes (other than the drug) that could on their own have caused the reaction?	−1	+2	0	2
6	Did the reaction reappear when a placebo was given?	−1	+1	0	0
7	Was the drug detected in the blood (or other fluid) in concentrations known to be toxic?	+1	0	0	0
8	Was the reaction more severe when the dose was increased, or less severe when the dose was decreased?	+1	0	0	0
9	Did the patient have a similar reaction to the same or similar drug in any previous exposure	+1	0	0	0
10	Was the adverse event confirmed by any objective evidence?	+1	0	0	0
				Total score	4

Probability Category: <0: Doubtful; 1–4: Possible; 5–8: Probable; >9: Definite Possible.

## Data Availability

The data that support the findings of this study, after adequate anonymization that protects the patient’s privacy, will be available on request from the corresponding authors (X.D.). The data are not publicly available due to them containing information that could compromise research participant privacy/consent.

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
