# Peer review of "COVID-19 Vaccine-Related Vogt–Koyanagi–Harada Disease Complicated by Central Serous Chorioretinopathy during Treatment Course: Case Report and Literature Review"

_vaccines, 2022, doi:10.3390/vaccines10111792_

Round 1
Reviewer 1 Report
My main suggestion is to emphasize the need for a check of potential side effects related to steroid therapy, especially in eyes with a thick choroid at baseline.
Even if this is not the first report about secondary VKH in subjects that received COVID19 vaccines, it should be remembered that it is a quite rare event.
---update
The most interesting aspect of this case-report is represented by the time sequence of the events, that is Covid-19 vaccine administration – intraocular inflammation (diagnosed as VKH) – steroid treatment – VKH resolution – secondary CSC – cyclosporin
Even if in the references other post-vaccine VKH cases have been already reported (ref 3,4,9, 12-15), authors should emphasize anyway that this is a rarity
My main suggestion is to shorten the discussion
Author Response
Thanks for your kind and important suggestions!
Yes, checking of potential side effects related to steroid therapy, especially in eyes with a pachychoroid is one of the key messages this case brought for us. We have emphasized this in Discussion part (line 219-222) and Conclusion part (line 246-249) as follows:
‘Thus, we consider that pachychoroid and systemic steroids might contribute together to the development of complicated CSC in our case. We recommend that ophthalmologists pay attention to this possible ocular side effects of systemic steroids, especially in eyes with a pachychoroid.’
‘However, systemic steroids may induce CSC, especially in a patient who has thick choroid. Here we reported a complicated CSC during the steroids treatment course of vaccine-related VKH and discussed the possible mechanisms of CSC onset. Although the acute CSC is self-limited and resolution could be observed with the tapering of steroids in most cases, physicians and ophthalmologists should be aware of this condition especially in eyes with a pachychoroid, distinguish complicated CSC with inflammation relapse carefully and adjust the medication timely.’
Secondly, we totally agree that post-vaccine VKH is a rarity. We have emphasized and added ‘VKH disease closely related to vaccines against SARS-CoV-2 is a quite rare event but has been recognized [3,4,9,12-15].’ in Discussion part Page 8 line 145-146.
Lastly, we have shortened the discussion and made it more concise.
Reviewer 2 Report
it is an interesting case report, well presented.
it should be specified what is the ocular condition of left eye after the followup visit at the end of july
Author Response
Thanks for your suggestion.
Quiet anterior and posterior segments were observed in his left eye after the follow-up visit at the end of July.
We have added this description in line 92-95: ‘However, in the following observation (2022-07-01 and 2022-07-29), the retinal fluid in his right eye was gradually increased (Figure 2b and 2c) without any sign of activated inflammation, while quiet anterior and posterior segments were observed in his left eye.’
Reviewer 3 Report
Both VKH disease and the CSC are the adverse event associate with covid-19 vaccination. The authors demonstrate a case clearly with VKH disease and 3months later onset CSC in this manuscript. I am curious that this VKH disease and the CSC are independent adverse events related to inactivate covid vaccination, or VKH disease with steroid use secondary CSC?
Author Response
Thanks for your important and interesting question. We have discussed it, too.
For one thing, according to recent studies, the reported time range between COVID-19 vaccination/infection and CSC onset is 24 hours to 1 month [1-6]. However, in our case, CSC (2022-07-01) developed more than 6 months after his third dose of vaccination (2021-11-24), which is much longer than ever reported cases.
For another thing, CSC is closely related with steroid use in this case. In our case, CSC developed after 6-month VKH management with steroids. More importantly, the subretinal fluid increased with addition dosage of prednisone and was completely absorbed after the tapering of oral prednisone by 5 mg a week. Therefore, we suggested that steroids work as a triggering factor in this case.
Moreover, we cannot exclude other risk factors for CSC in our case, such as pachychoroid. Previous study demonstrated pachychoroid as a risk factor in steroid-related CSC [7]. Although we did not know the central choroid thickness (CCT) of this patient before COVID-19 vaccination, CCT of his right eye was 393 μm when the choroid inflammation and edema OU was entirely relieved (2022-02-25).
To sum up, we do not consider CSC to be an independent adverse event directly related to inactivated-COVID vaccination in this case.
At the same time, we suggest that steroid treatment and pachychoroid contribute together to complicated CSC in our case. To avoid misunderstanding, we replaced the description of ‘secondary CSC’ with ‘steroid-related CSC’ and ‘complicated CSC’.
Thanks for your question again!
- 1. Sanjay S, Acharya I, Kawali A, Shetty R, Mahendradas PUnilateral recurrent central serous chorioretinopathy (CSCR) following COVID-19 vaccination- A multimodal imaging study. American journal of ophthalmology case reports 2022, 27:
- 2. Rama Raj P, Adler P, Chalasani R, Wan SAcute Unilateral Central Serous Chorioretinopathy after Immunization with Pfizer-BioNTech COVID-19 Vaccine: A Case Report and Literature Review. Seminars in ophthalmology 2022, 37 (6): 690-8.
- 3. Mechleb N, Khoueir Z, Assi ABilateral multifocal central serous retinopathy following mRNA COVID-19 vaccine. Journal francais d'ophtalmologie 2022, 45 (6): 603-7.
- 4. Hanhart J, Roditi E, Wasser L, Barhoum W, Zadok D, Brosh KCentral serous chorioretinopathy following the BNT162b2 mRNA vaccine. Journal francais d'ophtalmologie 2022, 45 (6): 597-602.
- 5. Bolletta E, Iannetta D, Mastrofilippo V, De Simone L, Gozzi F, Croci S, Bonacini M, Belloni L, Zerbini A, Adani C, Fontana L, Salvarani C, Cimino LUveitis and Other Ocular Complications Following COVID-19 Vaccination. Journal of clinical medicine 2021, 10 (24).
- 6. Mahjoub A, Dlensi A, Romdhane A, Ben Abdesslem N, Mahjoub A, Bachraoui C, Mahjoub H, Ghorbel M, Knani L, Krifa F[Bilateral central serous chorioretinopathy post-COVID-19]. Journal francais d'ophtalmologie 2021, 44 (10): 1484-90.
- 7. Han J, Hwang J, Kim J, Park K, Woo S. Changes in choroidal thickness after systemic administration of high-dose corticosteroids: a pilot study. Investigative ophthalmology & visual science 2014, 55 (1): 440-5.